# Footwear Affects Conventional and Sumo Deadlift Performance

**DOI:** 10.3390/sports9020027

**Published:** 2021-02-11

**Authors:** Kevin A. Valenzuela, Kellie A. Walters, Elizabeth L. Avila, Alexis S. Camacho, Fany Alvarado, Hunter J. Bennett

**Affiliations:** 1Movement Science Lab, Department of Kinesiology, California State University, Long Beach, CA 90840, USA; kellie.walters@csulb.edu (K.A.W.); Elizabeth.avila@student.csulb.edu (E.L.A.); camacho.alexis93@gmail.com (A.S.C.); fanyalvarado92@gmail.com (F.A.); 2Neuromechanics Lab, Old Dominion University, Norfolk, VA 23529, USA; hjbennet@odu.edu

**Keywords:** weight lifting, barefoot, shoes, resistance training

## Abstract

Barefoot weightlifting has become a popular training modality in recent years due to anecdotal suggestions of improved performance. However, research to support these anecdotal claims is limited. Therefore, the purpose of this study was to assess the differences between the conventional deadlift (CD) and the sumo deadlift (SD) in barefoot and shod conditions. On day one, one-repetition maximums (1 RM) were assessed for thirty subjects in both the CD and SD styles. At least 72 h later, subjects returned to perform five repetitions in four different conditions (barefoot and shod for both CD and SD) at 70% 1 RM. A 2 × 2 (footwear × lifting style) MANOVA was used to assess differences between peak vertical ground reaction force (VGRF), total mechanical work (WORK), barbell vertical displacement (DISP), peak vertical velocity (PV) and lift time (TIME) during the concentric phase. The CD displayed significant increases in VGRF, DISP, WORK, and TIME over the SD. The shod condition displayed increased WORK, DISP, and TIME compared to the barefoot condition. This study suggests that lifting barefoot does not improve performance as no differences in VGRF or PV were evident. The presence of a shoe does appear to increase the DISP and WORK required to complete the lift, suggesting an increased work load is present while wearing shoes.

## 1. Introduction

The deadlift is a closed chain exercise frequently utilized in strength and rehabilitation programs to develop posterior chain strength [1]. Two common styles include the sumo deadlift (SD) and conventional deadlift (CD). The primary difference between the styles lies in the technique of each lift. An SD traditionally has a wider stance, outturned feet, and hands are placed on the barbell medial to the knees. The CD has the lifter stand shoulder width apart and grasp the barbell just outside the knees [2]. Research seeking to determine which style elicits superior strength benefits is limited [2,3,4]. Due to the technical differences in the sumo and conventional style, the vertical displacement of the bar is greater in the conventional [5]. The SD’s wider stance and relative joint angles result in the barbell travelling a significantly shorter distance from lift off (LO) to lift completion (LC) [6,7]. Vertical displacement has a direct effect on mechanical work, where the CD had a significantly higher total work, while maintaining a relatively similar peak vertical ground reaction force (VGRF) in comparison to the SD [6,7,8]. Establishing the concept of mechanical work is important as it, in-part, dictates the amount of effort for a given exercise. Identifying the variable which alters the amount of mechanical work is also important as it establishes where the changes to the work occur, therefore exploring both the force levels and displacement is important.

The potential benefits of weight training while barefoot, in contrast to wearing shoes (shod), discussed in popular media are mostly theoretical. However, the effect of footwear has been researched in the context of several different unweighted activities and populations, including walking/running [9], postural stability [10], and in terms of how it relates to different sport maneuvers, such as jumping and landing [11]. Despite the growing body of research comparing barefoot, shod, and different shoe designs on locomotion, research concerning the effect of footwear on resistance training performance from a biomechanical perspective is quite limited. A study by Hammer et al., 2018 found that the rate of force development was reduced when wearing shoes compared to barefoot, which lends some support that there is a dissociation between the shoe (i.e., the sole) and ground. The same study also found that the medio-lateral center of pressure excursion was significantly greater in the shoe condition [12], which suggests that footwear could have a consequential effect on frontal plane joint mechanics. However, contrasting research has shown increased antero-posterior excursion while barefoot in comparison to shod conditions, when assessing postural control [10]. While the reduction in center of pressure excursion may provide some support for barefoot training, Hammer et al., 2018 found no significance in peak force or in time to peak force between the footwear conditions. The author indicates that the time to complete the concentric phase of the lift is similar in both conditions as well [12]. While the time aspect may be similar, the displacement would likely be smaller, thereby reducing the velocity at which the movement is performed. Lifting barefoot has been shown to reduce the bar velocity during a CD, and thus, it may decrease the power production, hindering the lifting performance while barefoot [13]. However, this has yet to be conclusively proven.

Maximizing deadlift performance through the lens of footwear choice has been a topic of debate that may also have an effect on strength gains [1,12]. Popular media (non-peer-reviewed sources) proposes that the benefits of barefoot training include increased balance and stability, a shorter range of motion, increased muscle recruitment, and a more efficient transfer of force [14,15]. Barefoot exercise can provide increased neurofeedback to improve stability whereas the soft sole would create instability [12]. With increased stability, rocking of the feet (i.e., shift in the center of pressure) would be minimized and there would be a greater recruitment of the hip extensors leading to a clean and efficient lift [15]. The shoe sole adds approximately 0.5–1.5 inches to the vertical displacement of the bar; therefore, without shoes, the lifter will have a shorter range of motion, have a smaller vertical displacement, and perform less work [15]. The shoe sole can be an unstable surface that also must compress before effectively transferring force, thereby creating a force delay. An extension of a delayed force transmission is that reduced VGRF is present, which would thereby negatively impact lifting performance when wearing shoes [12]. However, this has yet to be conclusively proven, as the majority of this evidence is anecdotal and not peer-reviewed scientific research.

Very few studies examining the biomechanical performance differences of footwear have been published and have yet to come to a consensus as to whether barefoot training is beneficial [1,12,13]. A recent review of the deadlifting literature has shown that less than half of all deadlift studies include female subjects [16]. To date, no studies have examined barefoot and shod lifting conditions for both the CD and SD collectively in the same sample population for a submaximal deadlift while including female subjects. The purpose of this study was to investigate differences of deadlifting styles and the effect of footwear on barbell displacement, total lift time, total mechanical work, peak concentric velocity, and peak VGRF during the concentric phase of the deadlift. It was hypothesized that the barefoot condition would result in less barbell vertical displacement and less total mechanical work, but no difference in lift time, peak velocity, or peak VGRF when compared to the shod condition of both the CD and the SD.

## 2. Materials and Methods

### 2.1. Subjects

Thirty subjects were recruited from university and the surrounding community (Table 1). The inclusion criteria required the subjects be between the ages of 18–35, had consistently performed the sumo or conventional type of deadlift in a strength training program of at least two days per week for six or more months, had no history of lower extremity or lower back injuries in the past six months and no history of lower extremity surgery. Informed consent was obtained from all subjects involved in the study. The study protocol was approved by the California State University, Long Beach Institutional Review Board on 17 May 2019 (IRB# 1418558-2). An a priori sample size calculation using G*Power (Version 3.1, Christian Albrechts University, Kiel, Germany) with an α of 0.05 and a power of 0.8 indicated a total sample size of 24 subjects was required.

### 2.2. Instrumentation

Reflective markers were attached to the ends of the barbell to measure the barbell trajectory. Raw marker data were collected using a 15-camera Qualisys Oqus 300 motion capture system (Qualisys North America, Inc. Buffalo Grove, IL, USA) sampling at 240 Hz. Two Bertec force plates (Bertec Corp., Columbus, Ohio, USA) embedded in the floor were used to collect raw force data, sampling at 1200 Hz. All subjects wore standardized running shoes (ASICS Gel-Flux 5, ASICS Corporation, Kobe, Japan) during the shod condition, which have a heel height of 29 mm and a heel drop of 10 mm for women’s sizes and 8 mm for men’s sizes according to the manufacturer’s specifications. All subjects used a standard 20 kg barbell, regardless of gender, mimicking availability of bar types in a traditional gym. No lifting aids (weight belts, straps, chalk, leg wraps, etc.) were permitted during 1 RM (one-repetition maximum) testing (which was used to obtain the 70% 1 RM value for the second day of testing) or during 70% 1 RM testing.

### 2.3. Procedures

On day one, subjects signed an informed consent and anthropometric data were collected (age, height, and mass; Table 1). Prior to both days of testing, subjects were instructed to avoid any exercise/weightlifting for a period of 24 h prior to the test date. A five-minute warm up was completed on a stationary bike at a self-selected pace. Subjects were also instructed to stretch as needed. After the completion of a warm-up, subjects’ one repetition maximum (1 RM) was determined using guidelines established by The National Strength and Conditioning Association. The subjects performed a light warm-up set consisting of 5–10 repetitions of the deadlift style (sumo or conventional) of their choosing, at 50% of their estimated 1 RM (self-reported), followed by a 1 min rest. Two heavier sets followed at 3–5 repetitions increasing by 10–20% for each set with a 2 min rest in between sets. Subjects then completed trials of one repetition with an increase of 10–20% of the load until failure to complete the repetition. After each successful 1 RM, the subject was given a 2–4 min rest. Once a failing lift was determined, a 2–4 min rest was observed and the load was decreased by 7–9 kg and repeated to determine the correct 1 RM. Following completion of the first 1 RM, a 10 min break was given [17] and then the 1 RM test was repeated for the other deadlift style (whichever was not performed during the first test). Following the completion of the second 1 RM, subjects were instructed to cool down and stretch at their own discretion. Including the warm up sets, lifters performed an average of five successful sets (plus one failed set) to find their 1 RM. Seventeen out of 30 lifters chose to perform the conventional deadlift first and the sumo deadlift second.

Day two was performed at least 72 h after the initial 1 RM testing to avoid any potential influence of fatigue [12]. Subjects began with a 5 min warm-up on the stationary bike at a self-selected pace. A warm up deadlift set of 8–10 repetitions at 50% of the subjects’ 1 RM was then completed. A 4 min rest period was then given while the weight on the barbell was adjusted and markers were attached to the ends of the barbell. Subjects completed four randomized sets (sumo and conventional with shoes and sumo and conventional without shoes) of five repetitions at 70% of their previously determined 1 RM. The only instructions provided to lifters were to not drop the bar back to the floor at the end of each concentric repetition and to perform the lift at a velocity they would utilize during a normal five-repetition set in their training. In the interest of time, both trials within each condition were completed sequentially. The lifts were completed at each lifter’s self-selected speed. A 5 min rest between each set was given to avoid possible fatigue.

### 2.4. Data Analysis

Raw marker data and force data were imported into a Visual 3D Biomechanical Suite (C-Motion Research Biomechanics, Germantown, MD, USA) for kinematic and kinetic variable computations. Raw marker and force data were filtered with a lowpass Butterworth filter with a cutoff frequency of 8 Hz. Data were assessed during the concentric phase of each repetition. The concentric phase was defined as the beginning of the ascent of the barbell from its lowest vertical position until the barbell reached its maximum vertical height. Mean values across the five repetitions of selected variables were used for statistical analysis for each person and condition. Variables of interest included peak VGRF (vertical ground reaction force ), peak vertical velocity of the barbell (PV, calculated as the first derivative of the barbell vertical position data), total displacement (DISP, measured as the difference in vertical position of the barbell from the ground to its peak height), total work (WORK, measured as the integral of the power curve based on the VGRF of the right leg and the velocity of the right side of the barbell), and total time (TIME) of the concentric phase of the lift.

### 2.5. Statistical Analysis

Data were assessed for outliers using the z-score and box-plot method which resulted in 17 outliers being removed. The Shapiro-Wilk test was used to determine normal distribution of data. The data violated Mauchly’s Test of Sphericity, and therefore, the Greenhouse Geisser method was used for data interpretation. A two-way repeated measure multivariate analysis of variance (MANOVA) was used to compare VGRF, PV, DISP, WORK, and TIME with and without shoes for the two deadlift styles. Post-hoc comparisons were made using a Bonferroni correction. Effect size was interpreted using partial η^2^, with an effect size of 0.1–0.6, 0.6–0.15, and ≥ 0.14 considered as a small, medium, and large effect, respectively. Data were analyzed using SPSS version 25 (IBM SPSS Statistics Inc., Chicago, IL, USA). Data were presented as means ± STD and significance levels were set at *p* < 0.05.

## 3. Results

### 3.1. Repeated Measures MANOVA

Mean values of all variables split by shoe condition, and deadlift type are presented in Table 2. There was no significant interaction between deadlift type and shoe condition for the MANOVA (*F*(5, 24) = 1.140, *p* = 0.367, Pillai’s Trace = 0.192, partial η^2^ = 0.192). However, there were significant main effects for shoe condition (*F*(5, 24) = 15.849, *p* < 0.001, Pillai’s Trace = 0.768, partial η^2^ = 0.768) and deadlift type (*F*(5, 24) = 29.011, *p* < 0.001, Pillai’s Trace = 0.858, partial η^2^ = 0.858) on the combined dependent variables.

### 3.2. Shoe Condition

There was a statistically significant effect on DISP (*F*(1, 28) = 57.948, *p* < 0.001, partial η^2^ = 0.674), WORK (*F* = 13.881, *p* = 0.001, partial η^2^ = 0.331), and TIME (*F* = 5.491, *p* <= 0.026, partial η^2^ = 0.164), but not for VGRF and PV. Wearing shoes while deadlifting resulted in significantly greater DISP (mean: 0.52 m (95% confidence intervals (CI): 0.51–0.54)) compared to barefoot (mean: 0.49 m (95% CI: 0.47–0.51), *p* < 0.001). WORK was also greater in shoes (mean: 418.32 J (95% CI: 396.50–440.14)) compared to barefoot (mean: 401.19 J (95% CI: 376.87–425.51), *p* = 0.001). Lastly, TIME was significantly increased while wearing shoes (mean: 1.06 s (95% CI: 1.02–1.10)) compared to barefoot (mean: 1.02 s (95% CI: 0.98–1.07), *p* = 0.026).

### 3.3. Deadlift Type

There was a statistically significant effect on VGRF (*F*(1, 28) = 13.558, *p* = 0.001, partial η^2^ = 0.326, DISP (*F* = 118.661, *p* < 0.001, partial η^2^ = 0.809, WORK (*F* = 57.177, *p* < 0.001, partial η^2^ = 0.671, and TIME (*F* = 10.802, *p =* 0.003, partial η^2^ = 0.278) but not on PV. The conventional deadlift resulted in a significantly greater VGRF (mean: 980.4 N (95%C CI: 924.6–1036.3)) compared to the sumo deadlift (mean: 938.2 N (95% CI: 889.8–986.5), *p* = 0.001). DISP was significantly greater in the CD (mean: 0.54 m (95%C CI: 0.52–0.55)) compared to the SD (mean: 0.48 m (95% CI: 0.46–0.50)) *p* < 0.001). WORK was significantly increased in CD (mean: 433.87 J (95%C CI: 410.10–457.65)) compared to SD (mean: 385.63 J (95% CI: 362.32–408.94), *p* < 0.001). Lastly, TIME was significantly increased in the SD (mean: 1.07 s (95%C CI: 1.02–1.12)) compared to the SD (mean: 1.01 s (95% CI: 0.97–1.05), *p* < 0.001).

## 4. Discussion

The purpose of this study was to assess the effects of footwear on the CD and the SD as regards the VGRF, PV, DISP, WORK, and TIME during the concentric phase. It was hypothesized that lifting barefoot would result in reduced DISP and WORK but have no effect on VGRF, PV, and TIME compared to wearing shoes. The results of this study partially confirm our hypotheses. While lifting barefoot, there was a decreased DISP of the barbell and WORK with no effect on VGRF or PV, which supports the hypotheses. However, there was also a decrease in TIME, which was not in agreement with the hypothesis. The CD also displayed increased VGRF, WORK, DISP, and TIME compared to the SD, regardless of footwear. There was no significant difference in PV between lift types.

Anecdotal evidence has suggested that barefoot lifting is better for grounding the foot (i.e., less center of pressure shift and more even distribution of pressure across the foot), maintaining three points of contact (heel, first metatarsal, and fifth metatarsal) which allows for increased force production. This study is not in agreement with the anecdotal argument as no differences were seen in peak VGRF while lifting barefoot at 70% 1 RM. This research agrees with previous scientific research which also did not show increased peak force during a barefoot deadlift [12]. Hammer et al., 2018 found no increases in peak force in the shod or barefoot condition at both 60 and 80% 1 RM of the CD [12]. The current study reported the same lack of differences in VGRF for 70% 1 RM in both the CD and SD. Hammer et al., 2018 did find an increased rate of force development in the barefoot condition; however, there was no difference in time to peak force [12]. The research by Hammer et al., 2018 showed that the shod condition displayed an increased medio-lateral center of pressure sway, indicating a more stable surface when barefoot. The authors suggested that this was potentially the result of the motor control systems achieving a steady state sooner, thus, allowing more effort to be focused on the force development [12], which could potentially explain the increased rate of force development while barefoot. Less energy would be directed towards achieving stability in the foot and more concentrated on the performance of the lift. However, as no increase in peak force is attained, being barefoot does not appear to help with maximal force development. The argument that could potentially be made here is with respect to efficiency of the movement. While barefoot, although no increase in VGRF is evident, less energy is wasted on controlling an unstable foot (i.e., increased center of pressure shifts while in shod conditions), thus making the movement more efficient as more energy can be directed toward the performance of the lift.

In comparison to a back squat, which has a similar triple extension movement pattern, deadlifts have been shown to have greater rates of force development than the back squat [18,19], which is beneficial for quick movements, but more VGRF is produced during the back squat [20]. Deciding which exercise to use depends on the overall goal desired. Peak power, on the other hand, may be a consideration for lifters in certain training programs. Power is a product of two vectors, force and velocity. Neither force or velocity were found to be significantly different in this research study, suggesting that being barefoot does not improve peak power either. Previous research has shown being barefoot does not have an effect on peak power during the CD at 60 and 80% 1 RM [12]. While power was not directly assessed here, given that there was no difference in peak velocity or VGRF between conditions, it is unlikely that there is a difference in peak power. A different study comparing the CD with the hex bar deadlift did show increased peak power in the hex bar at 30–80% 1 RM in conjunction with increased peak velocities but showed no differences in peak VGRF [1]. However, this power production may be impacted, as reduced bar velocity at knee pass has been shown during barefoot conditions during a CD [13]. For lifters striving to increase their power production, utilizing the hex bar may be the more appropriate method rather than the traditional bar, although the effects of shoes and barefoot lifting on the hex bar have yet to be assessed. An additional way that has been shown to improve power and velocity aspects is to add variable resistance to the lift [21], however, this results in decreases in peak VGRF [21,22].

While power, force, and velocity are critical components of performance, an important component of the barefoot argument is the amount of time spent under tension and the mechanical work accomplished as these are integral parts of the training process. WORK, DISP, and TIME all decreased while lifting barefoot. Removal of the shoe sole decreased the vertical displacement by approximately 0.03 m, resulting in 0.04 s less time under tension and 17.13 J less work per repetition per leg. This does not seem like a lot, but when considering the number of repetitions in a training session (or other interval of the training program), these numbers add up to a substantial amount of work lost over time. For example, if a lifter was performing a 5 × 5 (sets × repetitions) scheme, barefoot lifting would result in a reduction of 428.25 J per leg. Part of the goal of strength training programs for athletes is to utilize the amount of work that elicits the greatest benefit, which differs for different types of athletes. If the goal is maximal weight lifted, the reduced time under tension and displacement of the weight (resulting in less mechanical work) may indicate that lifting barefoot is the best option. However, the goal for many athletes is not to lift as much as possible in a single repetition, and therefore, utilizing shoes to increase the WORK performed may be of benefit as it forces the athlete to perform more mechanical work for every repetition. If time is a constraint, the shod condition may also allow the athlete to achieve the needed work quicker that a barefoot lift. Some lifters have taken to utilizing the barbell bounce, which also results in less work and impulse, but increased VGRF [23]. It is important to note that the maximal force produced while utilizing the bounce is developed later in the lift [23], so if a specific time period of peak force production is desired, utilizing the bounce should be exercised with caution.

In a comparison of the CD with the SD, it was found that mechanical work, vertical displacement, and peak VGRF were increased in the CD while lift time was reduced. The increased vertical displacement and mechanical work in the CD are congruent with past studies [6,7], as is the reduced vertical displacement in the SD [4,5]. The SD employs a wider stance width which moves the center of mass of the lifter closer to the ground, thereby making the distance from the beginning of the lift to full standing a shorter distance, leading to reduced vertical displacement. This study does differ in terms of time to completion, as past research has shown the time to completion for the CD and SD not to be different [5,6,7], while in this study, the CD was significantly longer than the SD (1.07 s and 1.02 s, respectively). Part of the determination of which form to use should be based on the desired goal of the training (maximal force production, peak velocity, etc.). This could also include target muscles. The SD has shown increased quadriceps (vastus medialis and lateralis) and tibialis anterior activity while the CD has shown increased medial gastrocnemius activity [24]. If lifters wish to target a certain muscle more than another, choosing the CD or the SD may impact this.

Lifter anthropometrics may affect the quality of these two deadlift styles, and therefore, may also affect performance. It has been suggested that lifters with shorter arms [3], and longer torsos [2] may be more appropriately suited for the SD, as better performance has been shown with respect to these characteristics. Lifters with increased height have shown a moderate correlation to vertical displacement of the bar during CD, while longer legs have been shown to be highly correlated to increased work [25]. These characteristics were not assessed in this study; however, males, on average, were 0.12m taller than females in this study (Table 1), which would likely result in an increased displacement of the bar. Additionally, males lifted heavier absolute loads by an average of >50 kg across both deadlift styles, which would result in greater VGRF and WORK. Past research has shown females to have increased average and peak concentric velocities at the same relative loads (≥90% 1 RM) compared to their male counterparts during a CD, but reduced average concentric velocity at 60–69% 1 RM during an SD [4]. It should be noted, however, that there is very little research concerning female weightlifting. A recent review of deadlift research indicated that less than half the articles (seven out of 19) included females in the sample population, with two of those studies being female only [16]. A great deal of the currently available results only apply to males, and thus, it is important to continue to add to the body of literature concerning female participation in weightlifting.

This study has some limitations. First, the study only examines the entire concentric phase of the lift. It is not subdivided into different phases as has been suggested by previous research [6]. It has been suggested that muscle activation patterns change (and therefore mechanics change) between concentric and eccentric phases so it would be beneficial to examine the eccentric phases in the future. Second, the tempo of the lift was not controlled during the concentric or eccentric phase. Subjects were instructed not to drop the bar between repetitions, however, there was no instruction about a pause or reset at the end of each repetition. The only other verbal instructions provided to the lifters were to mimic speeds they performed during a typical deadlift training session. Future research could consider examining the rate of force development and power-related variables in different speed conditions (e.g., fast as possible or a slow ascent). Third, the use of lifting aids was not allowed so it is unknown how items such as weight belts or lifting straps would impact these results. Fourth, the shoes used were standardized running shoes, not specific to lifting. Lifting shoes might provide different results as different heel heights could impact the variables analyzed. Fatigue during the 1 RM testing may have had an impact on the 1 RM performance as a rest period of ten minutes was provided between exercises, which may not have been sufficient to alleviate the fatigue. Finally, the VGRF and WORK values are for the right leg only. There may be differences between the legs, but given the exclusion criteria of injuries, it was assumed that these healthy lifters had symmetrical loading patterns.

## 5. Conclusions

Anecdotal evidence supporting barefoot lifting is largely unsubstantiated by the scientific community. However, this research does show that barefoot lifting could be considered more efficient than shod lifting, as reduced mechanical work (WORK) is required to complete the concentric phase of a deadlift. This appears to be largely due to the reduced vertical displacement (DISP) incurred during the lift, as no differences in VGRF were evident. Barefoot lifting does not appear to contribute to these performance variables and should therefore be used cautiously depending on the goals of the athlete.

## Figures and Tables

**Table 1 sports-09-00027-t001:** Anthropometric and 1 RM (one-repetition maximums) data (mean ± standard deviation).

Variable	Males (n = 16)	Females (n = 14)	Combined (n = 30)
Age (years)	24.9 ± 2.9	25.4 ± 3.4	25.1 ± 3.2
Height (m)	1.79 ± 0.09	1.67 ± 0.04	1.73 ± 0.10
Mass (kg)	88.00 ± 14.15	66.77 ± 5.64	78.09 ± 15.54
CD 1 RM (kg)	167.55 ± 40.44	105.14 ± 19.71	138.42 ± 45.74
SD 1 RM (kg)	155.36 ± 35.04	103.19 ± 18.26	131.01 ± 39.23

CD = conventional deadlift; SD = sumo deadlift; 1 RM = one repetition maximum.

**Table 2 sports-09-00027-t002:** Performance variables during conventional deadlift (CD) and sumo deadlift (SD) in shod and barefoot conditions (mean ± standard deviation).

	Conventional Deadlift		Sumo Deadlift	
Variable	Shod	Barefoot	Shod	Barefoot
VGRF (N) ^‡^	989.7 ± 230.2	993.9 ± 224.8	949.8 ± 212.9	949.1 ± 214.1
TIME (s) ^†,‡^	1.09 ± 0.13	1.05 ± 0.16	1.03 ± 0.11	0.99 ± 0.11
DISP (m) ^†,‡^	0.55 ± 0.04	0.52 ± 0.04	0.50 ± 0.05	0.46 ± 0.05
PV (m/s)	0.77 ± 0.10	0.77 ± 0.11	0.77 ± 0.11	0.73 ± 0.09
WORK (J) ^†,‡^	444.02 ± 99.52	433.97 ± 106.02	402.85 ± 100.72	378.42 ± 95.68

VGRF = peak vertical ground reaction force; TIME = time of concentric phase; DISP = vertical displacement of the barbell; PV = peak velocity of the barbell; WORK = integration of the force × velocity curve during the concentric phase; ^†^ significantly increased for shoe condition compared to barefoot condition; ^‡^ significantly increased for CD compared to SD.

## Data Availability

The data presented in this study are available on request from the corresponding author. The data are not publicly available due to privacy reasons.

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
