# Peer review of "Footwear Affects Conventional and Sumo Deadlift Performance"

_sports, 2021, doi:10.3390/sports9020027_

Round 1
Reviewer 1 Report
Thank you for opportunity for reviewing this interesting paper. The research adhere to reporting CONSORT guidelines. This paper provides useful information On effect of Footwear on Conventional and Sumo Deadlift Performance. If conducted with academic rigor, this article has the potential to be of value for physician and policymakers around the prevalence, cost and prevention of complications for athletes that use or not using footwear. Furthermore, in my opinion the topic and premise of the study would sit well within the journal to which it was submitted. The authors should be commended for undertaking this study.
Also, there are a major concerns with the manuscript that require attention prior to publication. These will be discussed below relative to the sections of the manuscript.
TITLE
The title of this manuscript are a little long. Perhaps a more concise version for clarity, interes and ease of read.
ABSTRACT
It is hard to get the detail in an abstract when the word count is limited and this is often the hardest part of a paper to write. However, I do feel that it would be beneficial to explain what specifically you are looking at in relation to footwear (this also applies to the main body of the paper). Is it the development of Sumo Deadlift Performance literature. This needs to be made clearer throughout the paper.
KEYWORDS:
Please use recognised MeSH terms as this will assist others when they are searching for information on your research topic. The following website will provide these (simply start typing in a keyword and see if it exists or find an alternative if it does not): https://www.ncbi.nlm.nih.gov/mesh
INTRODUCTION:
The introduction is weak. An introduction should announce your topic, provide context and a rationale for your work, while catching the reader´s interest and attention. The above has not been given in the introduction that I have read.
It is indeed important paper but it lacks several critical references, in which it was presented related with this importance of the footwear in the sports, and it should be emphasized in the INTRODUCTION or Discussion of the authors' paper. More info info in:
Effect of the cushioning running shoes in ground contact time of phases of gait https://pubmed.ncbi.nlm.nih.gov/30179793/
Stability of Three Different Sanitary Shoes on Healthcare Workers: A Cross-Sectional Study
https://www.mdpi.com/1660-4601/16/12/2126
Finally, please describe the hypothesis in this section.
MATERIAL AND METHODS:
This section is poor, needs to present a better rationale for the study and the methodology employed. Also, neither appear information related with inclusion and exclusion criteria, dates, protocol, and registered in clinical trials of this cuasi expermiental research.
Likewise more detail about information calculate sample size and data should be provided. Also, please need include the data and record code and all information related with the ethics committee and explain aspects ethics and legal requirement about this research.
RESULTS:
The results need provide clear results and describe them. Please include the table 1 in the the p-values in all test in this table
DISCUSSION:
In general the discussion of the results of the study is correct, authors describe the results, the limitations and compare with other researchs.
CONCLUSION:
These conclusions need to be softened, modified a in order to reflect only the study findings.
Reviewer 2 Report
Thank you for the opportunity to review this manuscript. This study investigated the effect of footwear on kinetic and kinematic variables obtained from summo and conventional deadlifts. This manuscript is well written and the results of this study would be of interest to practitioners prescribing exercise for general and athlete populations. Below I have detailed comments for the authors to consider.
Introduction
- Page 1 line 44 - The authors state that research concerning the effect of footwear from a biomechanical perspective is limited. It is recommend that the authors be specific and state that "research concerning the effect of footwear on resistance training performance from a biomechanical perspective is limited". There is a substantial body of literature on the effect of footwear on running from a biomechanical perspective.
- Page 2 line 66 - Recommended to change 'about' to 'approximately'
- Page 2 line 67 - The authors alternate between the terms 'lifter' and 'individual' throughout the manuscript. It is recommended that 'lifter' be used throughout.
Procedures
- What was the instruction to the lifter when completing the repetitions during the experimental conditions? Were the lifters instructed to lift the bar as quickly as possible? If measuring variables such as bar velocity , force and RFD the specific instructions provided to the lifter can significantly influence these results. It is mentioned in the limitations that lifters were instructed to not drop the bar, but were they instructed towards the "intention" of the lift?
- It is recommended that the authors provide a statement regarding how many attempts it generally took for the lifter to achieve their 1RM.
- Can the authors provided a rationale for completing both 1RM tests in the same day only separated by 10minutes. Fatigue may have impacted the results of the 2nd 1RM test.
- Lifters were able to choose which deadlift style they wanted to complete first. Information about how many chose to complete the CD or SD first. This important to determine if an order effect was present.
Round 2
Reviewer 1 Report
The authors has addressed all my comments and questions from previous report.